# Effectiveness and Safety of Molecular-Targeted Therapy after Nivolumab Plus Ipilimumab for Advanced or Metastatic Renal Cell Carcinoma: A Multicenter, Retrospective Cohort Study

**DOI:** 10.3390/cancers14194579

**Published:** 2022-09-21

**Authors:** Koji Iinuma, Koji Kameyama, Tomoki Taniguchi, Kei Kawada, Takashi Ishida, Kimiaki Takagi, Shingo Nagai, Torai Enomoto, Masayuki Tomioka, Makoto Kawase, Shinichi Takeuchi, Daiki Kato, Manabu Takai, Keita Nakane, Takuya Koie

**Affiliations:** 1Department of Urology, Graduate School of Medicine, Gifu University, Yanagido, Gifu 501-1194, Gifu, Japan; 2Department of Urology, Central Japan International Medical Center, 1-1 Kenkonomachi, Minokamo 505-8510, Gifu, Japan; 3Department of Urology, Ogaki Municipal Hospital, 4-86 Minaminokawa-cho, Ogaki 503-8502, Gifu, Japan; 4Department of Urology, Gifu Prefectural General Medical Center, 4-6-1 Noisiki, Gifu 500-8717, Gifu, Japan; 5Department of Urology, Gifu Municipal Hospital, 7-1 Kashimacho, Gifu 500-8513, Gifu, Japan; 6Department of Urology, Daiyukai Hospital, 1-9-9 Sakura, Ichinomiya 491-8551, Aichi, Japan; 7Department of Urology, Toyota Memorial Hospital, 1-1 Heiwacho, Toyota 471-8513, Aichi, Japan; 8Department of Urology, Matsunami General Hospital, 185-1 Kasamatsucho, Hashima-gun 501-6062, Gifu, Japan; 9Department of Urology, Japanese Red Cross Takayama Hospital, 3-113-11 Tenman-machi, Takayama 506-8550, Gifu, Japan

**Keywords:** nivolumab, ipilimumab, second-line treatment, molecular-targeted therapy, renal cell carcinoma, Japanese patients

## Abstract

**Simple Summary:**

We evaluated the efficacy and safety of molecular-targeted therapies (MTTs) in 29 patients who discontinued the combination therapy of nivolumab plus ipilimumab (NIVO+IPI) for advanced or metastatic renal cell carcinoma as real-world outcomes. Patients receiving MTTs had a median follow-up of 8 months. The objective response rate was 44.8%, and the disease control rate was 72.4%. After NIVO+IPI, the median overall survival was 18 months, and progression-free survival (PFS) was 8 months. Patients with bone metastases had a significantly shorter median PFS when treated with MTTs after NIVO+IPI than those without bone metastases (4 vs. 12 months, *p* = 0.012). MTTs may be a useful secondary treatment option after the discontinuation of NIVO+IPI.

**Abstract:**

This study aimed to evaluate the effectiveness and safety of molecular-targeted therapies (MTTs) after the discontinuation of nivolumab and ipilimumab (NIVO+IPI) combination therapy in patients who had been diagnosed with advanced/metastatic renal cell carcinoma as real-world outcomes. We enrolled patients treated with MTTs following initial therapy with NIVO+IPI at nine institutions in Japan. We evaluated the objective response rate (ORR) as the primary endpoint and disease control rate (DCR), best overall response, and oncological outcomes (overall survival (OS) and progression-free survival (PFS)) as the secondary endpoints. We also evaluated factors predictive of disease progression after the administration of MTTs. Patients were followed up for a median of 8 months. The ORR was 44.8%, and the DCR was 72.4%. The median OS and PFS of MTTs after NIVO+IPI were 18 months and 8 months, respectively. A total of 31% of patients experienced grade 3/4 MTT-related adverse events. The median PFS in patients with bone metastases was significantly shorter than that in those without bone metastases (4 vs. 12 months, *p* = 0.012). MTTs may be a useful secondary treatment option after the discontinuation of NIVO+IPI.

## 1. Introduction

The advent of immune checkpoint inhibitors (ICIs), such as nivolumab (NIVO; a programmed cell death protein 1 (PD-1) inhibitor) and ipilimumab (IPI; an anti-cytotoxic T-lymphocyte antigen 4 (CTLA-4) monoclonal antibody), has brought remarkable changes in the treatment strategies for advanced or metastatic renal cell carcinoma (mRCC) [1]. Recent randomized phase III trials (Checkmate 214 [2], KEYNOTE-426 [3], JAVELIN Renal 101 [4], IMmotion151 [5], CheckMate 9ER [6], and CLEAR [7]), demonstrating the combination of ICIs and/or molecular-targeted therapies (MTTs), revealed significant clinical advantages, especially in terms of oncological outcomes, compared with NIVO or MTT monotherapy in patients with mRCC [8,9]. Combination therapy with NIVO plus IPI (NIVO+IPI) is the only treatment for ICI combination therapy in Japan. In the CheckMate 214 trial, NIVO+IPI was compared with sunitinib (SUN) as first-line treatment for clear-cell mRCC [2]. Regarding the International Metastatic Renal Cell Carcinoma Database Consortium (IMDC), the median overall survival (OS) for mRCC patients with intermediate or poor risk was 47 months with NIVO+IPI and 26.6 months with SUN (*p* < 0.001) [10,11]. Although NIVO+IPI may have achieved a durable response for at least a 5-year follow-up period, 20% of patients with intermediate or poor risk regarding the IMDC risk classification developed progressive disease (PD) for their best overall response (BOR) [2]. In our previous study, the efficacy and safety of NIVO+IPI in patients with mRCC were evaluated using real-world data, and 23.5% of patients were noted to have developed PD [12]. Although combination ICI and/or MTT therapy has become the standard therapy for mRCC, treatment with ICIs might not confer equivalent clinical benefits in all patients with mRCC. Therefore, subsequent therapy after ICI regimens is important.

MTTs were the standard therapy for mRCC before the ICI era [13]. Current drugs approved for the treatment of mRCC in Japan are vascular endothelial growth factor tyrosine kinase receptor inhibitors ((TKIs) including sorafenib, SUN, axitinib (AXI), pazopanib, and cabozantinib (CABO)) and two mammalian targets of rapamycin inhibitors ((mTORis) including temsirolimus and everolimus). Even in the ICI era, MTTs have remained an important treatment strategy for mRCC and are recommended as a strategy after NIVO+IPI discontinuation [14,15]. Several studies reported the efficacy and safety of MTTs after ICIs for mRCC [16,17,18,19]. However, most of these studies analyzed patients with mRCC who received various ICI regimens. Therefore, the impact of MTTs after NIVO+IPI on real-world data remains unclear. A multicenter, retrospective cohort study was thus designed to evaluate the efficacy and safety of MTTs in patients who discontinued NIVO+IPI for mRCC as real-world outcomes. In addition, we evaluated the predictive factors for cancer progression after NIVO+IPI followed by MTTs.

## 2. Materials and Methods

### 2.1. Patients

This retrospective multicenter cohort study was carried out with nine institutions in Japan. Patients with mRCC who received NIVO+IPI followed by MTT between August 2018 and January 2022 were enrolled. All patients with mRCC had previously received NIVO+IPI as first-line treatment. Patients initially treated with NIVO+IPI were classified into intermediate- and poor-risk groups according to the IMDC risk classification. Patients treated with other systemic therapies, such as TKIs and mTORis, as the first-line approach and those with missing relevant data were excluded from the study. The clinical characteristics of patients receiving MTTs after NIVO+IPI, namely, age, sex, Eastern Cooperative Oncology Group performance status (ECOG-PS) [20], histology, the reason for discontinuation of NIVO+IPI, the presence or absence of surgery after NIVO+IPI, metastatic site, and the number of metastases, were collected.

This study was approved by the institutional review board of Gifu University (authorization number: 2020-271). This was a retrospective study; there was no role for consent and prospective enrollment. Patients received standard-of-care second-line therapy and were retrospectively evaluated. According to the provisions of the ethics committee and ethics guidelines in Japan, study information is disclosed to the public in the case of retrospective and/or observational studies, with materials such as existing documentation. Details of this study are available at https://www.med.gifu-u.ac.jp/visitors/disclosure/docs/2020-271.pdf, accessed on 3 March 2020.

### 2.2. Subsequent Therapy after Discontinuation of NIVO+IPI

All patients in this analysis received MTTs, including SUN, AXI, and CABO, as subsequent therapies. The drugs used were determined by treatment institutions. Treatment was continued until radiological disease progression or the development of unacceptable toxicity for treatment-related adverse events (TRAEs).

### 2.3. Patient Evaluation

Patients were evaluated at baseline, prior to the implementation of MTTs, based on their complete history and physical examination, as well as chest, abdominal, and pelvic computed tomography (CT) and magnetic resonance imaging (MRI) findings. American Joint Committee on Cancer Staging Manual Tumor staging was used to determine the tumor stage [21].

The outcomes of all patients were evaluated using CT or MRI, performed at 1–3 month intervals until radiological disease progression or treatment discontinuation due to TRAEs. According to the Response Evaluation Criteria in Solid Tumors (RECIST) guidelines version 1.1 [22], the BOR was recorded as complete response (CR), partial response (PR), stable disease (SD), or PD. The objective response rate (ORR) and the disease control rate (DCR) were defined as the proportion of patients with RECIST-based BOR of CR or PR and the proportion of patients with RECIST-based CR, PR, or SD, respectively.

### 2.4. Safety

According to the National Cancer Institute Common Terminology Criteria for Adverse Events (version 5.0) [23], TRAEs were evaluated between the date of the first MTT and at least 100 days after the last MTT.

### 2.5. Statistical Analysis

The primary endpoint was the ORR. The secondary endpoints were DCR, BOR, OS, and progression-free survival (PFS). Data were analyzed using software JMP 14 (SAS Institute Inc., Cary, NC, USA). The follow-up period ranged from the date of the first MTT to the last follow-up examination or the date of confirmed death. OS was defined as the duration from the first MTT following NIVO+IPI discontinuation to the date of all-cause death. PFS was defined as the interval from the initiation of MTT after the discontinuation of NIVO+IPI owing to disease progression. In addition, the efficiency of the initial MTTs for discontinuing of NIVO+IPI was investigated and divided into four subgroups as follows: IMDC risk classification, reasons for discontinuation of NIVO+IPI, MTT regimens after the discontinuation of NIVO+IPI, and with or without bone metastases. OS and PFS were evaluated using the Kaplan–Meier method. Differences in clinical variables were evaluated using the log-rank test. All two-sided *p*-values of < 0.05 were considered statistically significant.

## 3. Results

### 3.1. Patients

Between August 2018 and January 2022, 61 patients with mRCC were treated with NIVO+IPI, and 29 patients (47.5%) were treated with MTTs after the discontinuation of NIVO+IPI at nine institutions in Japan (Figure 1). The demographic data of the enrolled patients are presented in Table 1. Seven patients (24.1%) had ECOG-PS ≤ 2. The major metastatic sites were the lungs and lymph nodes. Bone metastases occurred in 11 patients (37.9%). The median follow-up period from the initiation of MTTs after the discontinuation of NIVO+IPI to the date of analysis or death was 8 months (interquartile range (IQR): 2.0–16.5).

### 3.2. First MTT after Discontinuation of NIVO+IPI

The first MTT after the discontinuation of NIVO+IPI is shown in Table 2. In this study, axitinib was the most commonly used drug in the MTTs.

### 3.3. Efficacy and Oncological Outcomes

The treatment effects in patients who underwent MTTs after NIVO+IPI are listed in Table 3. The median OS and PFS from the initiation of MTTs after NIVO+IPI were 18 months (95% confidence intervals (CI): 8.0–not reached (NR)) and 8.0 months (95% CI: 4.0–12.0), respectively (Figure 2a,b). The subgroup analyses of OS and PFS are shown in Figure 3. The median PFS of MTTs after NIVO+IPI was significantly shorter in patients with bone metastases than in those without bone metastases (4.0 vs. 12.0 months, *p* = 0.012) (Figure 3h).

### 3.4. Safety

TRAEs are presented in Table 4. Major grade 3–4 TRAEs were hypertension (13.8%) and palmar–plantar erythrodysesthesia syndrome (6.9%). At the end of the follow-up period, none of the patients discontinued MTTs regarding TRAEs, and none of the patients had treatment-related death.

## 4. Discussion

Currently, the treatment strategy using ICIs for mRCC has become the standard care, according to several guidelines [24,25]. In addition to a minimum 5-year follow-up period in the Checkmate214 trial, NIVO+IPI demonstrated long-term efficacy benefits as compared with SUN in patients with mRCC [10]. Although long-term effects can be expected in patients who benefit from NIVO+IPI, 50% of patients who had been diagnosed with intermediate and poor risk according to the IMDC risk stratification showed disease progression at 12 months [10]. The establishment of optimal sequential therapies after NIVO+IPI may play an important role in achieving long-term survival in patients with mRCC.

A small number of studies have reported that MTTs have favorable antitumor activity and safety after ICIs [16,17,18,19]. Auvray et al. reported the efficacy and safety in 33 patients with mRCC who received subsequent TKIs after NIVO+IPI [16]. The ORR was 36%, and the DCR was 76% [16]. The median PFS and OS at 12 months were 8 months and 54%, respectively [16]. A total of 42.4% of patients had grade 3–4 TRAEs [16]. Shah et al. evaluated subsequent MTTs after first-line ICI treatments, including ICI monotherapy, ICI combination therapy, and ICIs with MTTs [17]. The ORR and the DCR were 41.2% and 94.1%, respectively [17]. The median PFS was 13 months, and OS at 12 months was 79.6% [17]. In patients who were initially administered NIVO+IPI, the ORR, the DCR, the median PFS, and OS at 12 months were 43.8%, 93.8%, 11.9 months, and 81%, respectively [17]. Owing to TRAEs, 27% of patients discontinued MTTs [17]. Tomita et al. revealed the efficacy and safety of MTTs after NIVO+IPI in 19 Japanese patients [18]. The ORR was 32%, and the DCR was 84% [18]. The median PFS and OS at 12 months were 16.3 months and 89.5%, respectively [18]. A total of 51% of patients had Grade 3/4 TRAEs [18]. These reports were for patients after clinical trials and included patients with mRCC who were classified into the favorable-risk group according to the IMDC. To the best of our knowledge, this study is the first to analyze real-world data to provide evidence of the efficacy and safety of MTTs after NIVO+IPI in Japan. The ORR and the DCR were 44.8% and 72.4%, respectively, and one patient (3.4%) achieved CR in this study. The median PFS and OS rates at 12 months were 8.0 months and 61.6%, respectively. Although the proportion of patients with mRCC who were diagnosed in the poor-risk group based on the IMDC was relatively higher than that in previous studies, oncological outcomes were comparable. As for the safety, nine patients (31%) experienced grade 3/4 TRAEs, and no patients discontinued MTTs owing to TRAE. The safety of ICI treatment followed by MTTs was similar to that in previous studies.

Several studies have reported that mRCC patients with bone metastases have significantly poor oncological outcomes [24,25]. Dong et al. reported a significant survival advantage for patients with mRCC without bone metastases compared with those with bone metastases (*p* < 0.0001) [26]. Beuselinck et al. studied 223 patients treated with SUN for mRCC to determine whether the presence of bone metastases affected their outcomes [27]. With a median follow-up of 40 months, patients with bone metastases had significantly shorter median PFS and OS than those without bone metastases (8.2 vs. 19.1 months, *p* < 0.0001, and 19.5 vs. 38.5 months, *p* < 0.0001, respectively) [25]. This study also revealed that the median PFS of patients treated with MTTs after NIVO+IPI was significantly reduced in the presence of bone metastases compared with the absence of bone metastases (4 vs. 12 months, *p* = 0.012). Therefore, bone metastasis was an independent prognostic factor in patients treated with MTTs after ICI therapy for mRCC. 

A subgroup analysis in the Checkmate214 trial showed that NIVO+IPI had significantly better oncological outcomes than SUN in patients without bone metastases, whereas NIVO+IPI tended to be better than SUN in patients with bone metastases [2]. However, there were no significant associations with these outcomes [2]. According to the Checkmate 9ER trial, the combination of nivolumab and cabozantinib therapy was significantly better at PFS than SUN in patients with bone metastases [6]. Therefore, the choice of this regimen as first-line therapy may be important for patients with bone metastases.

Our study had several limitations. First, this was a retrospective multicenter cohort study. Therefore, this study might have a selection bias due to diagnostic and therapeutic variations across participating institutions. Second, it had a relatively small sample size and a relatively short follow-up period. Finally, the use of bone resorption inhibitors may also have been considered in patients with bone metastases even though there are no data on the use of bone resorption inhibitors in this study.

## 5. Conclusions

Although a relatively small number of participants were enrolled in this multicenter retrospective study, the oncological outcomes and TRAE profiles were equivalent to those of other studies. Therefore, MTTs as second-line therapy may have potential advantages and lead to a treatment effect on mRCC patients treated with NIVO+IPI. This is the current clinical paradigm, and the retrospective study confirmed the clinical activity of MTT in this setting. Further research is warranted to evaluate the long-term outcomes of MTTs.

## Figures and Tables

**Figure 1 cancers-14-04579-f001:**
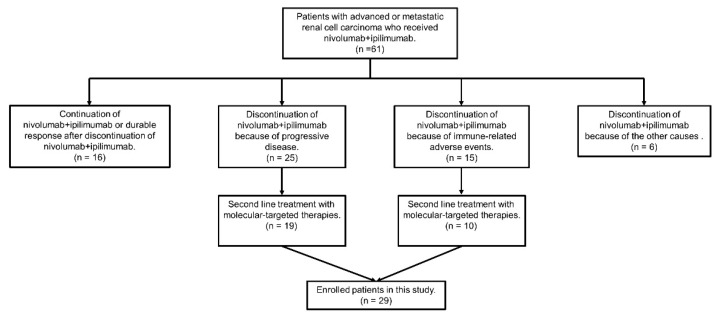
Flowchart of study population selection.

**Figure 2 cancers-14-04579-f002:**
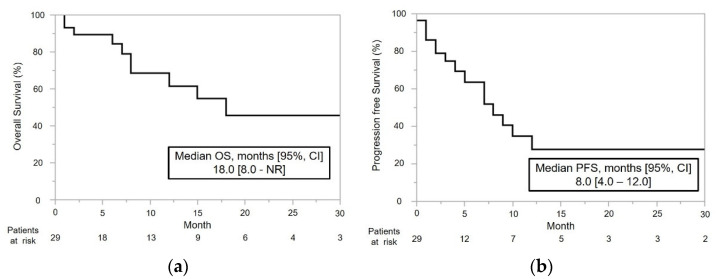
Overall survival (OS) and progression-free survival (PFS) from the date of the first molecular-targeted therapy in patients who discontinued nivolumab plus ipilimumab combination therapy. The median overall survival (OS) and progression-free survival (PFS) were (**a**) 18 months and (**b**) 8 months, respectively.

**Figure 3 cancers-14-04579-f003:**
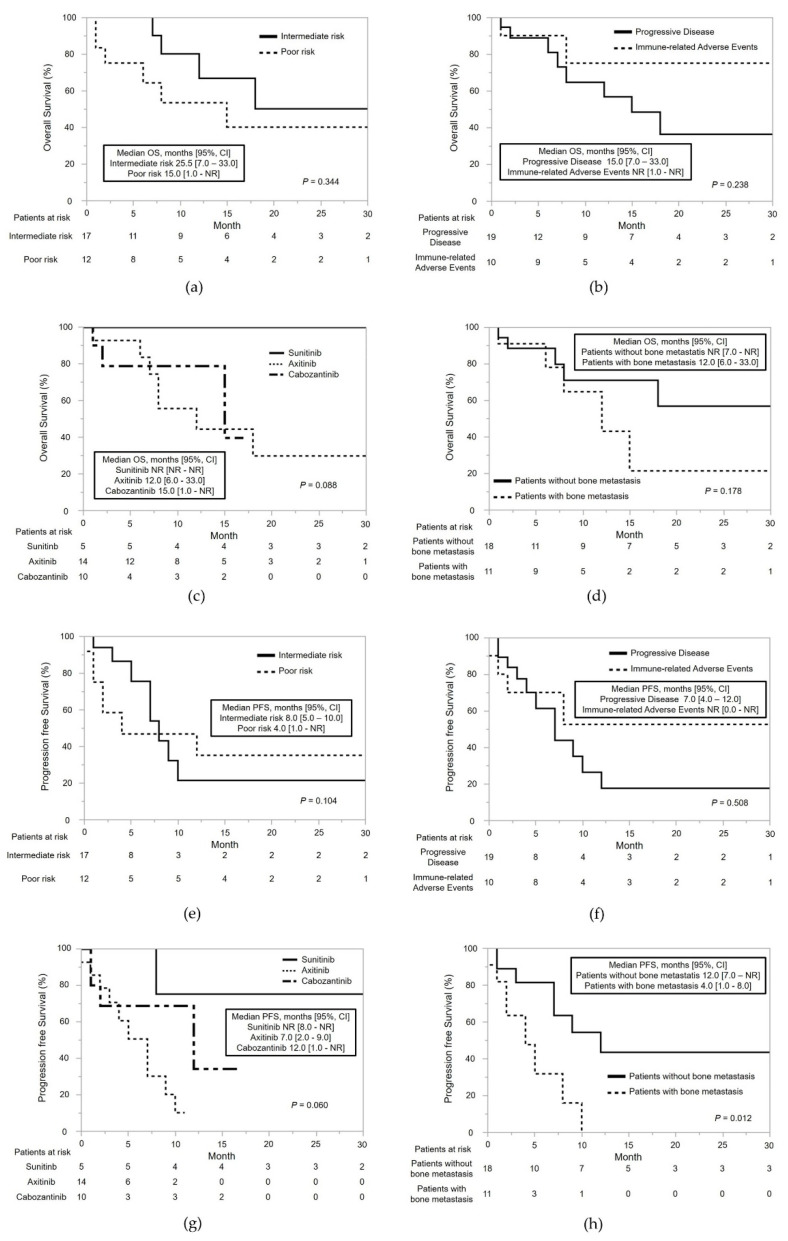
Overall survival (OS) and progression-free survival (PFS) of patients treated with molecular-targeted therapies (MTTs) after nivolumab plus ipilimumab combination therapy (NIVO+IPI). (**a**) OS from the date of the first MTT in patients who discontinued NIVO+IPI for intermediate/poor risk metastatic renal cell carcinoma (mRCC), as determined according to the International Metastatic Renal Cell Carcinoma Database Consortium risk classification. (**b**) OS from the date of the first MTT after NIVO+IPI in patients stratified according to the reason for the NIVO+IPI discontinuation (disease progression or adverse events). (**c**) OS from the date of the first MTT after NIVO+IPI in patients stratified according to MTT regimen (sunitinib, axitinib, or cabozantinib). (**d**) OS from the date of the first MTT after NIVO+IPI in patients stratified by the presence or absence of bone metastases from renal cell carcinoma. (**e**) PFS from the date of the first MTT after NIVO+IPI in patients with intermediate/poor-risk mRCC, as determined according to the IMDC risk classification. (**f**) PFS from the date of the first MTT after NIVO+IPI in patients stratified according to the reason for NIVO+IPI discontinuation (disease progression or adverse events). (**g**) PFS from the initiation of MTT after discontinuation of NIVO+IPI, stratified by MTT regimens and sunitinib, axitinib, or cabozantinib. (**h**) PFS from the initiation of MTT after discontinuation of NIVO+IPI, stratified by with or without bone metastases.

**Table 1 cancers-14-04579-t001:** Patient demographics at the time of the first molecular-targeted therapy in patients who discontinued nivolumab and ipilimumab combination therapy.

Covariate	
Age (years; median, interquartile range)	68.0 (55.5–74.0)
Gender (number, %)	
Male	24 (82.8)
Female	5 (17.2)
Eastern Cooperative Oncology Group performance status (number, %)	
0	12 (41.4)
1	10 (34.5)
2	5 (17.2)
3	2 (6.9)
Primary IMDC risk classification (number, %)	
Intermediate	17 (58.6)
Poor	12 (41.4)
Histology	
Clear cell renal cell carcinoma	20 (69.0)
Papillary renal cell carcinoma	1 (3.4)
Xp11.2 translocation carcinomas	1 (3.4)
Unknown	7 (24.2)
Reason for discontinuation of NIVO+IPI (number, %)	
Progression disease	19 (65.5)
Adverse events	10 (34.5)
Patients who underwent surgery after the administration of NIVO+IPI (number, %)	1 (3.4)
Number of metastatic sites	
0	1 (3.4)
1	4 (13.8)
2	17 (58.7)
≥3	7 (24.1)
Total number of metastatic sites (number, %)	
Lung	20 (69.0)
Lymph node	13 (44.8)
Bone	11 (37.9)
Liver	5 (17.2)
Adrenal gland	4 (13.8)
Pancreas	1 (3.4)
Others	6 (20.7)

IMDC, International Metastatic Renal Cell Carcinoma Database Consortium; NIVO+IPI, nivolumab plus ipilimumab combination therapy.

**Table 2 cancers-14-04579-t002:** First molecular-targeted therapy in patients who discontinued nivolumab plus ipilimumab combination therapy.

Targeted Therapy (*n*, %)	Total (*n* = 29)
Axitinib	14 (48.3)
Cabozantinib	10 (34.5)
Sunitinib	5 (17.2)

**Table 3 cancers-14-04579-t003:** Response to targeted therapy after discontinuation of nivolumab plus ipilimumab combination therapy.

	Total (*n* = 29)
Objective response rate (CR + PR; number, %)	13 (44.8)
Disease control rate (CR + PR + SD; number, %)	10 (72.4)
Best overall response (number, %)	
CR	1 (3.4)
PR	12 (41.4)
SD	8 (27.6)
PD	8 (27.6)

CI, confidence interval; CR, complete response; PD, progressive disease; PR, partial response; SD, stable disease.

**Table 4 cancers-14-04579-t004:** Adverse events with targeted therapy after nivolumab and ipilimumab.

Event (Number, %)	Any Grade	Grade 3/4
Treatment-related adverse events	20 (69.0)	9 (31.0)
Palmar–plantar erythrodysesthesia syndrome	7 (24.1)	2 (6.9)
Hypertension	6 (20.7)	4 (13.8)
Increased AST	5 (17.2)	0
Increased ALT	5 (17.2)	0
Hypothyroidism	4 (13.8)	0
Diarrhea	3 (10.3)	0
Erythema multiforme	3 (10.3)	0
Decreased white blood cells	1 (3.4)	1 (3.4)
Anorexia	1 (3.4)	1 (3.4)
Purpura	1 (3.4)	1 (3.4)
Proteinuria	1 (3.4)	1 (3.4)
Cholecystitis	1 (3.4)	0

ALT, alanine aminotransferase; AST, aspartate aminotransferase.

## Data Availability

Data and materials are provided in this paper.

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
