# Peer review of "Effectiveness and Safety of Molecular-Targeted Therapy after Nivolumab Plus Ipilimumab for Advanced or Metastatic Renal Cell Carcinoma: A Multicenter, Retrospective Cohort Study"

_cancers, 2022, doi:10.3390/cancers14194579_

Round 1

Reviewer 1 Report

Post immunotherapy, several retrospective studies and sub analyses of prospective trials have been conducted to review the role of TKI's in mRCC.  These studies indicate no concerning significance or unusual toxicity.  Post immunotherapy, retrospective studies as well as subgroup analysis of respective nonrandomized trials suggest TKI such as cabozantinib and axitinib are considered safe. It is well-known that an advanced renal cell carcinoma, osseous metastases are associated with poor prognosis.

This is a retrospective study of mRCC patients who received frontline dual checkpoint inhibitor therapy and were on second line treatments with TKI's.  The authors conducted a retrospective chart review of 29 patients previously treated with frontline ipilimumab and nivolumab for metastatic renal cell carcinoma and subsequently received second line therapy with axitinib, cabozantinib or sunitinib. The primary end point-  Objective response rate was 44.8% and disease control rate was 72%.   Bone metastases were associated with a shorter PFS with second line therapy with tyrosine kinase inhibition, 4 months versus 12 months in patients without bone metastases.

I have a few comments and suggestions for the authors:

The authors note in the abstract that the aim of their studies to evaluate the “effectiveness and safety of targeted therapies after ipilimumab and nivolumab”. Further, they evaluate the prognostic impact of osseous metastases in these immunotherapy pretreated patients. It would be more appropriate to use “real world outcomes” that were evaluated retrospectively in this paper.

In the abstract section, and

38, please change the word safeness to safety.
  Further, since this was a retrospective study, there was no role for consent and prospective enrollment.  Patients received standard of care second line therapy and were evaluated retrospectively.

In the conclusion section, the authors note that targeted therapy second in therapy may have potential advantages and lead to treatment effect for MR CC patients treated with nivolumab nivolumab (lines 270-71).  I think this should be rephrased to state that this is the current clinical paradigm and the retrospective study confirms clinical activity of targeted therapy in the setting.

In line 272-bone metastases ARE a significant protective factor for disease progression

Author Response

7, September 2022

Dr. Samuel C. Mok

Editor-in-Chief

Cancers

Dear Editor:

Thank you very much for the review of our manuscript titled “Bone metastasis as a predictive factor for disease progression after nivolumab plus ipilimumab followed by targeted therapy for advanced or metastatic renal cell carcinoma: A multicenter, retrospective cohort study.”

We sincerely appreciate all valuable comments and suggestions, which helped us to improve the quality of our manuscript. Our responses to the Reviewers’ comments are described below in a point-to-point manner. Appropriate changes, suggested by the Reviewers, have been introduced to the manuscript (track-changes mode in the red color font). Let me emphasize our full readiness to make any further improvements to the manuscript.

We hope that our manuscript will be acceptable for publication in the Cancers.

We look forward to hearing from you.

Yours sincerely,

Takuya Koie

Department of Urology

Gifu University Graduate School of Medicine

1-1 Yanagido, Gifu, Gifu 501-1194, Japan

TEL.: +81-582-30-6338

FAX: +81-582-30-6341

e-mail: goodwin@gifu-u.ac.jp

Responses to the reviewer's comments

We would like to thank the Reviewers for taking the time and effort necessary to review the manuscript. We sincerely appreciate all the valuable comments and suggestions, which helped us to improve the quality of the manuscript.

Response to Reviewer 1

The authors appreciate the reviewer’s comments. The authors’ point-by-point responses to the comments are given below.

  1. It would be more appropriate to use “real world outcomes” that were evaluated retrospectively in this paper.

Response:

We have added “real-world outcomes” on lines 31, 41, and 90.

  1. In the abstract section, and 38, please change the word safeness to

safety.

Response:

We have changed the word “safeness” to “safety” on line 39.

  1. Further, since this was a retrospective study, there was no role for consent and prospective enrollment. Patients received standard of care second line therapy and were evaluated retrospectively.

Response:

We have revised the following sentences on line 106:

This was a retrospective study; there was no role for consent and prospective enrollment. Patients received standard of care second-line therapy and were evaluated retrospectively The requirement for patient consent was waived owing to the retrospective study design.

  1. In the conclusion section, the authors note that targeted therapy second

in therapy may have potential advantages and lead to treatment effect for MR CC patients treated with nivolumab (lines 270-71).  I think this should be rephrased to state that this is the current clinical paradigm and the retrospective study confirms clinical activity of targeted therapy in the setting.

Response:

We have added the following sentence on line 289:

This is the current clinical paradigm, and the retrospective study confirms the clinical activity of MTT in this setting.

  1. In line 272-bone metastases ARE a significant protective factor for disease progression

Response:

We have revised the following sentence on line 291:

Bone metastasis is a significant predictive factor for disease progression and requires careful follow-up during the administration of MTTs.

Reviewer 2 Report

Revision of the following manuscript:

 BONE METASTASIS AS A PREDICTIVE FACTOR FOR DISEASE PROGRESSION AFTER NIVOLUMAB PLUS IPILIMUMAB FOLLOWED BY TARGETED THERAPY FOR ADVANCED OR METASTATIC RENAL CELL CARCINOMA:

A MULTICENTER, RETROSPECTIVE COHORT STUDY.

Although this paper reports the results of a retrospective analysis, we point out that the

the topic is of great clinical interest for oncologists and urologists.

We should like to underline the following points, and would appreciate it if you would consider them in your initial editorial evaluation:

  1. The object examined is of great current interest since the impact of molecular-targeted therapies (TTs) after nivolumab plus ipilimumab for advanced or metastatic renal cell cancer on real-world data remains unclear.
  2. The treatment with immune check-point inhibitor (ICIs) might not confer equivalent clinical benefits in all patients with advanced or metastatic renal cell carcinoma (mRCC); therefore, the evaluation of efficacy and safety of subsequent therapy after ICI regiment is important
  3. Despite the unquestionable importance of the subject, the report presents some critical issues (some of which are however already mentioned in the discussion by the authors) that require careful analysis:
    1. The study is a retrospective analysis with the well known limitations
    2. the number of patients analized was relatively low (29 out 61 enrolled in the study);
    3. similarly, the median follow-up was rather limited (8 months);
    4. as regards the objective response rate (ORR), the results are consistent with literature data (44.8%);
    5. due to short follow-up time, it is difficult to deduce the actual meaning of the reported median OS (mOS) and median PFS (mPFS),18 months and 8.0 months respectively;
    6. mPFS was shorter for patients with bone mets (4 vs 12 mos); however there were only 11 patients in this subgroup, so the observation, even if consistent with literature data (27 bibliography), does not allow to make a strong conclusion about  the presence of bone metastases as an independent prognostic factor.
    7. It would also have been important to consider the possible use of bone resorption inhibitors in the group with bone metastases.
    8. the rate of discontinuation of treatment regarding treatment related adverse events (TRAEs) was 31%; this rate does not differ significantly from the literature.
  4. In the end, considering all the above comments, I believe that the focus of the  paper is the sequence data (ICIsàTTs) in real world setting instead of  the prognostic significance of bone metastases.

Author Response

10, September, 2022

Dr. Samuel C. Mok

Editor-in-Chief

Cancers

Dear Editor:

Thank you very much for the review of our manuscript titled “Bone metastasis as a predictive factor for disease progression after nivolumab plus ipilimumab followed by targeted therapy for advanced or metastatic renal cell carcinoma: A multicenter, retrospective cohort study.”

We sincerely appreciate all valuable comments and suggestions, which helped us to improve the quality of our manuscript. Our responses to the Reviewers’ comments are described below in a point-to-point manner. Appropriate changes, suggested by the Reviewers, have been introduced to the manuscript (track-changes mode in the red color font). Let me emphasize our full readiness to make any further improvements to the manuscript.

We hope that our manuscript will be acceptable for publication in the Cancers.

We look forward to hearing from you.

Yours sincerely,

Takuya Koie

Department of Urology

Gifu University Graduate School of Medicine

1-1 Yanagido, Gifu, Gifu 501-1194, Japan

TEL.: +81-582-30-6338

FAX: +81-582-30-6341

e-mail: goodwin@gifu-u.ac.jp

Responses to the reviewer's comments

We would like to thank the Reviewers for taking the time and effort necessary to review the manuscript. We sincerely appreciate all the valuable comments and suggestions, which helped us to improve the quality of the manuscript.

Response to Reviewer 2

The authors appreciate the reviewer’s comments. The authors’ point-by-point responses to the comments are given below.

  1. The object examined is of great current interest since the impact of molecular-targeted therapies (TTs) after nivolumab plus ipilimumab for advanced or metastatic renal cell cancer on real-world data remains unclear.

Response:

The authors agree with reviewer’s comments.

  1. The treatment with immune check-point inhibitor (ICIs) might not confer equivalent clinical benefits in all patients with advanced or metastatic renal cell carcinoma (mRCC); therefore, the evaluation of efficacy and safety of subsequent therapy after ICI regiment is important

Response:

The authors agree with reviewer’s comments.

  1. Despite the unquestionable importance of the subject, the report presents some critical issues (some of which are however already mentioned in the discussion by the authors) that require careful analysis:
  2. The study is a retrospective analysis with the well-known limitations
  3. the number of patients analyzed was relatively low (29 out 61 enrolled in the study);
  4. similarly, the median follow-up was rather limited (8 months);
  5. as regards the objective response rate (ORR), the results are consistent with literature data (44.8%);
  6. due to short follow-up time, it is difficult to deduce the actual meaning of the reported median OS (mOS) and median PFS (mPFS),18 months and 8.0 months respectively;
  7. mPFS was shorter for patients with bone mets (4 vs 12 mos); however there were only 11 patients in this subgroup, so the observation, even if consistent with literature data (27 bibliography), does not allow to make a strong conclusion about the presence of bone metastases as an independent prognostic factor.
  8. It would also have been important to consider the possible use of bone resorption inhibitors in the group with bone metastases.
  9. the rate of discontinuation of treatment regarding treatment related adverse events (TRAEs) was 31%; this rate does not differ significantly from the literature.

Response:

We have added the limitation that there are no data about bone resorption inhibitors in this study on line 280.

Finally, the use of bone resorption inhibitors may also be considered in patients with bone metastases even though there are no data on the use of bone resorption inhibitors in this study.

  1. In the end, considering all the above comments, I believe that the focus of the paper is the sequence data (ICIsàTTs) in real world setting instead of the prognostic significance of bone metastases.

Response:

We have modified the focus to be on sequence data after ICI.

We have revised the Title on line 2.

The effectiveness and safety of molecular-targeted therapy Bone metastasis as

a predictive factor for disease progression after nivolumab plus ipilimumab

followed by targeted therapy for advanced or metastatic renal cell carcinoma: A

multicenter, retrospective cohort study

We have revised the following sentence on line 38.

Although MTTs may be a useful secondary treatment option after the

discontinuation of NIVO+IPI, bone metastasis was signif-icantly associated with

poor oncological outcomes for MTTs after NIVO+IPI.

We have revised the following sentence on line 52.

Although MTTs may be a useful secondary treatment option after the

discontinuation of NIVO+IPI, bone metastasis was significantly associated with

poor oncological outcomes for MTTs after NIVO+IPI.

We have deleted the following sentence on line 289.

Bone metastasis is a significant protective factor for disease progression and

requires careful follow-up during the administration of MTTs.

Reviewer 3 Report

This is a report of a multicenter retrospective cohort study analyzing prognostic factors in metastatic renal cell carcinoma treated with molecular targeted therapy after nivolumab plus ipilimumab combination therapy. Although the analysis is limited to 29 patients, the limited number of patients is unavoidable considering the population that received the specific therapy.

#1. I would think it is not common to abbreviate molecular-targeted therapy as TT. Please reconsider.

#2. The text in Figures 1 and 3 is unclear and difficult to decipher. Please revise the figures.

#3. As stated by the authors, bone metastasis has been reported to be a poor prognostic factor in metastatic renal cell carcinoma. This study is unique in that the analysis was limited to patients who received nivolumab plus ipilimumab followed by molecular targeted therapy, and it is necessary to consider whether the poor prognostic factor of bone metastasis was related to the treatment regimen. Are nivolumab plus ipilimumab combination therapy and molecular targeted therapies less likely to be effective in the presence of bone metastases? Please add your considerations by re-reviewing previous reports.

Author Response

7, September 2022

Dr. Samuel C. Mok

Editor-in-Chief

Cancers

Dear Editor:

Thank you very much for the review of our manuscript titled “Bone metastasis as a predictive factor for disease progression after nivolumab plus ipilimumab followed by targeted therapy for advanced or metastatic renal cell carcinoma: A multicenter, retrospective cohort study.”

We sincerely appreciate all valuable comments and suggestions, which helped us to improve the quality of our manuscript. Our responses to the Reviewers’ comments are described below in a point-to-point manner. Appropriate changes, suggested by the Reviewers, have been introduced to the manuscript (track-changes mode in the red color font). Let me emphasize our full readiness to make any further improvements to the manuscript.

We hope that our manuscript will be acceptable for publication in the Cancers.

We look forward to hearing from you.

Yours sincerely,

Takuya Koie

Department of Urology

Gifu University Graduate School of Medicine

1-1 Yanagido, Gifu, Gifu 501-1194, Japan

TEL.: +81-582-30-6338

FAX: +81-582-30-6341

e-mail: goodwin@gifu-u.ac.jp

Responses to the reviewer's comments

We would like to thank the Reviewers for taking the time and effort necessary to review the manuscript. We sincerely appreciate all the valuable comments and suggestions, which helped us to improve the quality of the manuscript.

Response to Reviewer 2

The authors appreciate the reviewer’s comments. The authors’ point-by-point responses to the comments are given below.

  1. I would think it is not common to abbreviate molecular-targeted therapy

as TT. Please reconsider.

Response:

We have changed the abbreviation “TT” to “MTT.”

  1. The text in Figures 1 and 3 is unclear and difficult to decipher. Please

revise the figures.

Response:

We have revised Figures 1 and 3.

  1. As stated by the authors, bone metastasis has been reported to be a

poor prognostic factor in metastatic renal cell carcinoma. This study is unique in that the analysis was limited to patients who received nivolumab plus ipilimumab followed by molecular targeted therapy, and it is necessary to consider whether the poor prognostic factor of bone metastasis was related to the treatment regimen. Are nivolumab plus ipilimumab combination therapy and molecular targeted therapies less likely to be effective in the presence of bone metastases? Please add your considerations by re-reviewing previous reports.

Response:

We have added following sentences on line 273:

Subgroup analysis in the Checkmate214 trial showed that NIVO+IPI had significantly better oncological outcomes than SUN in patients without bone metastases, whereas NIVO+IPI tended to be better than SUN in patients with bone metastases [2]. However, there was no significant association with these outcomes [2]. According to the Checkmate 9ER trial, the combination of nivolumab and cabozantinib therapy was significantly better at PFS than SUN in patients with bone metastases [6]. Therefore, the choice of this regimen as first-line therapy may be important for patients with bone metastases.

Round 2

Reviewer 2 Report

I do not have further comments